Optimizing 5G network performance with dynamic resource allocation, robust encryption and Quality of Service (QoS) enhancement

http://orcid.org/0000-0002-8682-6609 Alashjaee Abdullah M. 1
http://orcid.org/0000-0002-3830-1736 Kushwaha Sumit 2 sumit.kushwaha1@gmail.com
Alamro Hayam 3
Hassan Asma Abbas 4
Alanazi Fuhid 5
Mohamed Abdullah 6
1 Department of Computer Sciences, Faculty of Computing and Information Technology, Northern Border University , Rafha , Saudi Arabia
2 Department of Computer Applications, University Institute of Computing, Chandigarh University , Mohali, Punjab , India
3 Department of Information Systems, College of Computer and Information Sciences, Princess Nourah bint Abdulrahman University , Riyad , Saudi Arabia
4 Department of Computer Science, Community College-Girls Section, King Khalid University , Aseer , Saudi Arabia
5 Department of Information Systems, Faculty of Computer and Information Systems, Islamic University of Madinah , Medina , Saudi Arabia
6 Research Centre, Future University in Egypt , New Cairo , Egypt
Alatas Bilal
Electronic publication date: 2024 Nov 29
Publication date: 2024
Volume: 10
Electronic Location ID: e2567
Received 2024 May 30; Accepted 2024 Nov 8
Copyright: © 2024 Alashjaee et al.
Copyright year: 2024
Copyright holder: Alashjaee et al.
License: This is an open access article distributed under the terms of the Creative Commons Attribution License, which permits unrestricted use, distribution, reproduction and adaptation in any medium and for any purpose provided that it is properly attributed. For attribution, the original author(s), title, publication source (PeerJ Computer Science) and either DOI or URL of the article must be cited.
License URL: https://creativecommons.org/licenses/by/4.0/

Keywords: Optimization, VoIP, MCM, QoS, Digital life

Funding: King Khalid University RGP2/319/45 Princess Nourah Bint Abdulrahman University, Riyadh, Saudi Arabia PNURSP2024R361 Northern Border University, Arar, KSA NBU-FFR-2024-1576-07 Future University in Egypt (FUE) This work was supported by the Deanship of Research and Graduate Studies at King Khalid University through Large Research Project under grant number RGP2/319/45. Princess Nourah bint Abdulrahman University Researchers Supporting Project number (PNURSP2024R361), Princess Nourah bint Abdulrahman University, Riyadh, Saudi Arabia. This work was also supported by the Deanship of Scientific Research at Northern Border University, Arar, KSA for funding this research work through the project number “NBU-FFR-2024-1576-07”. This work is also funded by the Future University in Egypt (FUE). There was no additional external funding received for this work. The funders had no role in study design, data collection and analysis, decision to publish, or preparation of the manuscript.

==============================
The International Telecommunication Union (ITU) predicts a substantial and swift increase in global mobile data traffic. The predictions suggest that this growth will vary from 390 EB (exabytes) to 5,016 EB (exabytes) from 2024 to 2030, accordingly. This work presents a new maximum capacity model (MCM) to improve the dynamic resource allocation, robust encryption, and Quality of Service (QoS) in 5G networks which helps to meet the growing need for high-bandwidth applications such as Voice over Internet Protocol (VoIP) and video streaming. Our proposed MCM model enhances data transmission by employing dynamic resource allocation, prioritised traffic management, and robust end-to-end encryption techniques, thereby guaranteeing efficient and safe data delivery. The encryption procedure is applied to the header cypher, while the output parameters of the payload are altered. This indicates that only the sender and recipient will possess exclusive knowledge of the final outcome. In result, the comparative analyses clearly show that the MCM model outperforms over conventional models in terms of QoS packet planner, QoS packet scheduler, standard packet selection, traffic management, maximum data rate, and bandwidth utilisation.

Introduction

The introduction of the fifth generation (5G) of wireless technology represents a major advancement in mobile communications. It provides unparalleled speeds and reliability, enabling support for current mobile broadband applications as well as emerging technologies like the Internet of Things (IoT), industrial automation, smart cities, and autonomous vehicles. Nevertheless, these technological breakthroughs bring about intricate difficulties in network administration, specifically in upholding the Quality of Service (QoS) that customers anticipate amidst the significant rise in data quantities and connection density (Popovski et al., 2018; Kushwaha, 2023; Kumar & Kushwaha, 2024; Yuvaraj, Karthikeyan & Praghash, 2021).

The shift from 4G to 5G entails more than a mere augmentation in velocity and capacity. It represents a fundamental change towards networks that are more adaptable, expandable, and effective in serving a wider range of services with strict performance demands. Unlike the fourth generation (4G) of wireless technology, which mostly focuses on establishing connectivity, the fifth generation (5G) intends to ensure specific service levels, enhance user experience, and improve operational performance in situations with a very large number of devices and new service paradigms (Kumar & Kushwaha, 2023; Wang, Wang & Wang, 2019; Wang et al., 2020).

Key architectural improvements in 5G compared to previous generations include the implementation of sophisticated antenna technologies like Massive MIMO, enhanced frequency efficiency, and the utilisation of greater spectrum bands in both sub-6 GHz and mmWave ranges. In addition, 5G technology incorporates network slicing, enabling operators to establish several virtual networks that have unique quality and use attributes. This feature is particularly advantageous for meeting the varying needs of various applications in a manner that is both cost-effective and adaptable (Kousar et al., 2023; Hannah et al., 2022; Xi & Qixuan, 2018; Song et al., 2017).

QoS in network management pertains to the network’s capacity to transmit traffic with minimal latency and optimal dependability. Within the realm of 5G, the importance of QoS is heightened due to the crucial nature of certain applications it facilitates. For example, in the field of telemedicine, even a slight delay of milliseconds can have a significant impact on the results of remote surgical procedures. In the context of autonomous driving, the transmission of data in real-time is of utmost importance for ensuring the safety and efficiency of the cars (Praghash & Karthikeyan, 2021; Sultan & El Sayed, 2023).

In order to successfully handle QoS, 5G networks employ advanced techniques such as deep packet inspection (DPI), traffic shaping, and prioritisation. These techniques enable intelligent management of network traffic by identifying, categorising, and giving priority to data packets. These strategies guarantee that essential services such as emergency communications are not impeded by less important services.

Although 5G possesses strong capabilities, there are various obstacles that impede its ability to provide ideal QoS as (Singh et al., 2023; Syed et al., 2022): Scalability: The huge amount of data and the multitude of connected devices in 5G networks necessitate highly scalable solutions that can adapt to changing network conditions without compromising service quality.

Security and privacy are major concerns for 5G networks due to the growing number of access points and reliance on virtualization. These networks are susceptible to many security threats that, if not effectively addressed, can negatively impact the QoS.

Interference management is particularly important in densely populated urban areas, where the presence of several devices can have a considerable negative impact on signal quality and, consequently, the QoS. Efficient solutions for managing interference are crucial for optimising network performance.

Resource allocation is a challenging task that involves efficiently distributing resources among several devices and services, each with varying QoS requirements. Conventional static resource allocation strategies are impractical in 5G networks because of their dynamic nature.

In order to tackle these difficulties, this study suggests a maximum capacity model (MCM) that integrates dynamic resource allocation, advanced traffic prioritisation algorithms, and strong security measures to improve QoS in 5G networks. The model possesses a range of talents, which include (Kumar & Kushwaha, 2022; Kushwaha, 2022): Dynamic resource allocation in MCM differs from static allocation by adapting bandwidth and other resources in real-time according to demand, usage patterns, and priority levels, rather than remaining fixed regardless of changing network conditions.

Intelligent traffic prioritisation: MCM employs machine learning algorithms to forecast traffic trends and effectively prioritise data flow. This ensures that key applications receive the required resources promptly, without any delay.

MCM incorporates sophisticated encryption standards and real-time security processes to ensure the protection of data integrity and privacy, while maintaining high speed and service quality.

Network slicing technology allows MCM to develop tailored virtual networks that effectively and efficiently suit the specific QoS needs of many applications, ranging from IoT to high-speed mobile internet.

Furthermore, improving QoS in 5G networks has significant consequences (Kumar & Kushwaha, 2023): Improved QoS has a positive economic impact by increasing the dependability of services. This, in turn, creates new prospects and business models, especially in sectors such as telemedicine, entertainment, and transportation.

Enhanced network performance can have a substantial social impact by greatly improving user experience, resulting in smoother and more immersive interactions with technology.

Technical impact: The implementation of strong QoS systems promotes innovation in various technical domains by guaranteeing the dependability and effectiveness of fundamental communication infrastructures.

As 5G networks become increasingly integral to supporting next-generation applications, the need for advanced QoS mechanisms becomes imperative. The MCM suggested provides a holistic solution to the difficulties encountered by existing technologies, offering a scalable, secure, and efficient approach to network traffic management and improvement of overall service quality.

The research gap identified in the article revolves around the dynamic and scalable characteristics of 5G networks, which are not fully addressed by existing QoS models. The motivation for the work stems from the challenges in maintaining high QoS in 5G networks due to the increase in connected devices, dynamic traffic, and the need for secure data transmission. The proposed MCM aims to address these challenges by integrating dynamic resource allocation, intelligent traffic prioritization, and robust security measures to improve QoS.

The article is organized into several sections, each focusing on different aspects of the MCM and its comparison with existing models. The ‘Introduction’ section highlights the model’s ability to improve dynamic resource allocation, robust encryption, and QoS. The ‘Literature Review’ section provides a comprehensive review of the current state of 5G technology, its challenges, and the importance of QoS in this context. The ‘Proposed MCM Model’ section details its steps for information encryption and decryption, and its focus on efficient resource utilization and network management. The ‘Results and Discussions’ section discusses the results of comparing the MCM model with other models. The ‘Conclusion’ section concludes with a summary of the MCM model’s benefits and its potential impact on network performance, as well as suggestions for future research directions.

Literature review

5G technology has rapidly evolved, bringing about a new era of digital communication. This era is defined by extremely fast speeds, reduced latency, and enhanced connectivity. Nevertheless, the incorporation of QoS in such a dynamic setting poses intricate difficulties. This literature review examines different techniques and models suggested in recent research to improve QoS in 5G networks, with a particular focus on the use of novel solutions such as the MCM to effectively tackle these difficulties.

5G technology has the goal of ensuring widespread internet access, facilitating the connection of billions of devices, and offering extremely fast response times and great dependability, which are crucial for vital applications (Tam et al., 2024). 5G is unique in that it needs to accommodate a wide range of services and demands, including fast mobile internet, large-scale machine-type communications (mMTC), and highly reliable low-latency communications (URLLC) (Zhou, Islam & Chang, 2023).

QoS, in this sense, pertains to the network’s capacity to ensure specific performance requirements for the transmission of data or traffic throughout the network. Parameters commonly encompass metrics such as data transfer speeds, total delay from source to destination, and variations in delay, among other factors. Ensuring QoS in 5G entails effectively controlling many parameters to fulfil the specific demands of different applications, especially during periods of high network traffic (Ahmad et al., 2018).

Multiple variables contribute to the intricacy of continually offering service of superior quality in 5G networks: Network density and heterogeneity pose challenges in maintaining QoS due to the growing number and variety of devices connected to the network. The diversity of network resources, stemming from a multitude of data sources and varying spectral bands, introduces an additional level of intricacy (Ahmad et al., 2017).

The dynamic nature of 5G traffic, which is affected by user movement and changing network conditions, requires adaptive QoS methods. Traditional static QoS regulations are inadequate to handle these demands (Kushwaha, 2012).

Resource allocation: Effectively distributing finite network resources, like as bandwidth and energy, is essential for preserving service quality, particularly in densely populated network areas. Efficient network management requires the prioritisation of traffic and the dynamic allocation of resources (Kushwaha, Kumar & Jain, 2011).

Concerns regarding the protection of data and personal information: Given the sensitive nature of the data being carried, it is of utmost importance to prioritise the security and privacy of communications inside 5G networks. Nevertheless, security measures frequently clash with QoS objectives because to their potential to cause delays and incur more overhead (Kushwaha, 2023).

Several methodologies and models have been suggested to tackle the distinct issues of QoS management in 5G networks: Network slicing is a method that entails establishing numerous virtual networks, known as slices, on a solitary physical network infrastructure. Every slice is independent and possesses its own distinct network architecture, allowing for customisation to accommodate the unique requirements of various applications. Network slicing improves QoS by providing greater control over the allocation of resources and management of traffic (Kumar et al., 2023).

Deep packet inspection (DPI) is a technique used to analyse and control network traffic in a detailed manner. It allows for more efficient prioritisation of traffic. DPI, or Deep Packet Inspection, enables the analysis of the data portion of a packet. This allows for the implementation of advanced traffic management strategies that can adaptively prioritise based on the current state of the network and QoS requirements (Yuvaraj, Karthikeyan & Praghash, 2021).

Using machine learning techniques for predictive analysis can greatly improve the management of QoS. These algorithms can proactively modify resource allocation and traffic management parameters to ensure service quality by anticipating traffic loads and anticipated network problems (Cao et al., 2019).

Resource prioritisation protocols, such as IEEE 802.11e and others, aim to improve QoS by giving priority to different types of traffic, such as voice over IP and streaming video. The purpose of these protocols is to guarantee that essential services are allocated sufficient bandwidth and minimal latency to operate effectively (Li, Ota & Dong, 2018).

Although the current solutions offer frameworks for QoS management, they frequently fail to completely tackle the extremely dynamic and scalable characteristics of 5G networks. The majority of models are ineffective in properly incorporating deep learning to accurately forecast and adjust to dynamic network conditions in real-time. Moreover, there is a substantial demand for models that can effectively manage the trade-offs between stringent security/privacy requirements and the necessity for fast, low-delay communications.

The article introduces the MCM as a solution to address the existing gaps. The MCM is a scalable and dynamic resource allocation engine that utilises real-time data analytics and machine learning to optimise traffic flow and resource distribution. This model not only satisfies the requirement for optimal efficiency and adaptability in distributing resources, but also improves security measures without sacrificing the QoS.

Proposed mcm model

The proposed MCM model is depicted below in Fig. 1.

Figure 1 Proposed maximum capacity model block diagram.

Step 1: Information encryption: While encryption enables us to hide the contents of information, it is still possible to transmit information to a specific individual (or an authorized individual on their behalf). The manager lacks comprehension of the material, nonetheless, it is indisputable that we exchanged information on a particular day and at a specified hour. Interacting with certain folks might often draw unwelcome scrutiny.

Step 2: Information decryption: It is necessary for all devices linked to the network to have a robust transmission with each other. These devices are performing their tasks with efficiency and dependability, ensuring that the current telecommunications infrastructure remains unaffected. Furthermore, the components that form the basis of information sharing further augment its velocity and analysis.

Step 3: Device security: In order to safeguard the data, it is important to do an analysis of the devices that utilise it. Malfunctions that are present in those devices can provide security risks if they are not thoroughly examined. Consequently, this leads to a decrease in credibility, thereby strengthening the need for protection. It is crucial to take this into account since it might have a significant impact on all devices connected to the network.

Network equipment manufacturers face the challenge of creating novel software and hardware combinations to satisfy the growing bandwidth requirements of applications. In the networking sector, it is frequently seen that seemingly identical versions of equipment from the same manufacturer are really customised to address specific issues. For instance, most Ethernet switch manufacturers also provide conventional switches utilised in enterprises for building storage networks and coordinating operator services. Even among models with similar prices, there may be slight variations in setup that make them particularly well-suited for specific jobs. This technique is specifically developed to remove extraneous data that is not relevant to a particular user. Therefore, an alternative format is synonymous with the phrase “Save Information” in this particular situation. Figure 2 illustrates the complete range of potential MCM stages. Dual storage queue: A dual storage queue is an effective method for achieving efficient access when entering by maintaining two steps.

Digital key: A USB port type key is required to securely store and access information on devices using a unique code.

Security test: By examining the individuals accessing your conversations via an application, we can confirm the safety of this platform for exchanging images.

Figure 2 Proposed MCM model potential stages.

The diagram in Fig. 3 illustrates the system for allocating and managing QoS. The range of technologies that a hardware device can support should be a crucial factor in the decision-making process. Figure 3 demonstrates that specific transportation operations and categories can be carried out entirely in hardware, hence eliminating the requirement for the computer’s central processing unit and primary memory. This is because the software-level handling of traffic from other apps significantly limits both the overall and maximum performance. Due to their complex hardware setup, multilayer switches have the capability to consistently send IP packets, even when all of their ports are being utilised.

Figure 3 QoS allocation and management system.

Furthermore, these switches, particularly the more affordable options, are not suitable for our needs due to their lack of support for the required protocols. Additionally, if we intend to utilise more advanced encapsulation methods such as generic routing encapsulation (GRE) or multiprotocol label switching (MPLS) establishing a connection using such switches would be at best impractical. Affordable central processing unit. Therefore, in order to tackle these problems, it is necessary to use software rather than routers and hardware that rely on a powerful central CPU. At its highest level of performance, this product offers support for a wide number of protocols and technologies that are not accessible on switches of similar price. The following method, Algorithm 1, represents a model that has the highest possible capacity.

Algorithm 1 Proposed maximum capacity model (MCM).

1.    Start	
2.    Initialize the input details	
3.    if (Resource = available)	
4.      Then move to utilization	
5.      Update the details in database	
6.         if (Resource = not available)	
7.            Then move to user to waiting list	
8.            if (waiting time > allotted time)	
9.            Then declare min QoS	
10.           else if (data validation ≤ allot time)	
11.           Then declare max QoS	
12.    End	

Manufacturers often include two maximum performance measurements in their product documentation: one in bytes per second and the other in bits per second. Parsing packet data on most network devices costs a disproportionate amount of time and energy. The device is required to receive the packet, determine the appropriate transmission route, create a new header, and, if needed, send the packet. Undoubtedly, the number of packets is more relevant than the data rate in this particular situation. If two streams with the same velocity but different packet sizes are flowing, the stream with the smaller packet size is expected to have higher performance. It is important to consider this if the network will be used for IP telephony, as the actual performance in terms of bits per second is far lower than what is stated.

Estimating the strain on equipment resources can be challenging when managing a combination of transit and additional services, such as NAT and VPN. Equipment makers or their partners often do load testing on many models in different settings and publish the findings on the internet as comparison tables. Addressing these discoveries greatly streamlines the process of choosing the most suitable model.

Results and discussions

The proposed MCM model has been compared against the Existing End-To-End Security Framework (EESF) (Bugeja et al., 2019), Fast Authentication and Data Transfer Method (FADT) (Cao et al., 2019), Efficient Resource Scheduling (ERS) (Yu et al., 2023), and an Efficient Multi-Factor User Authentication (EMUA) (Abdi et al., 2024).

Quality of service packet planner

The third component of QoS that we must understand is QoS packet planning. In essence, the main responsibility of a QoS packet planner is to formulate traffic designs. In order to accomplish this, the packet planner collects packets from various rows and subsequently labels these packets with preferences and flow rate, as described in Fig. 4.

Figure 4 Comparison of QoS packet planner.

Due to the manufacturer’s provision of its own administration interface, it is not feasible to provide a comprehensive description of how QoS is enabled on all routers and carrier-class network equipment. Typically, accessing the Device Management page involves entering its URL (usually 192.168.1.1) in a web browser, logging in as the administrator using the credentials provided in the user manual, and navigating to the website.

QoS packet scheduler

The QoS packet on TP-Link routers may be found within the Schedule Bandwidth Control menu. To activate this feature, select the option “Enable bandwidth control” and specify the maximum speed for both incoming and outgoing traffic. An IP address or address range must be provided. Additionally, please indicate the specific types of ports and packet transfers to which this rule is applicable. TP-Link has implemented a novel graphical representation of the administrative panel, which is now standard on all newly released models. The QoS scheduling feature may be found in the “Additional settings” part of the “Data Priority” section, as depicted in Fig. 5.

Figure 5 Comparison of QoS packet scheduler.

There is a wide variety of routers available, including those designed for home and office use, as well as more advanced carrier-grade equipment. None of them possess the QoS function, and if they do, the way it is implemented may differ in the available configuration options. DPI systems have the ability to prioritise specific devices, emphasise specific types of traffic such as video or voice, identify applications that do not have pre-existing titles and data structures, and modify the priority field of packets to enforce QoS rules. We are unable to provide a comprehensive guide for setting up each individual device, however we can outline the fundamental steps to initiate the use of the QoS function in order to enhance the web browsing experience.

Standard packet selection

Classification of packets in general results in the formation of streams. As mentioned before, a notable feature of Floss Peck is the nature of its service. Assigns the thread’s priority based on the service type. The generic packet classifier is tasked with detecting the service of the failed particle, as shown in Fig. 6, and subsequently categorising the packets accordingly.

Figure 6 Comparison of standard packet selection.

The Internet speed for a telecommunications operator refers to the uplink or access speed provided by numerous providers. This value is calculated and allocated to all subscribers based on their payment plans. In order to guarantee customer satisfaction with the service provided, it is necessary to address the optimization and appropriate delivery of the service through QoS guidelines. It is crucial to ascertain the true speed of the home internet before configuring QoS, as it frequently does not align with the speed claimed by the provider. We need to establish definitions for outgoing and incoming velocities.

Traffic management

An important concern is the inability to prioritise network traffic based on its originating machine. Multiple programmes, and even operating systems, can live on a single platform, each generating its own traffic. Figure 7 demonstrates the need to assign separate priority to each award. For instance, while the most efficient distribution method may be ideal for one application, another may necessitate a higher amount of bandwidth. In this instance, the Traffic Control API is employed. The Traffic Control API is a programming interface that enables us to modify the quality of service for certain packets. The API’s primary use is to control the QoS for different traffic groups.

Figure 7 Comparison of traffic management.

Maximum data rate

The limit is a predetermined value that is determined by the outstanding package parameter. Once the number of packets that have been violated exceeds a specific threshold, the QoS will decline any further allocation of bandwidth for the network adapter until the number of violated packets falls below the maximum permissible limit. The set timer resolution option determines the smallest unit of time (measured in microseconds) that the QoS packet planner uses to schedule packets. Crucially, this mechanism regulates the highest possible rate at which packets can be stored in a queue for delivery, as depicted in Fig. 8.

Figure 8 Comparison of maximum data rate.

Prior to commencing the device configuration, it is imperative that we establish precise objectives for designing QoS. When setting up a home router, it is important to prioritise the working computer over other devices with internet access. This will enable smooth work and uninterrupted video streaming for online gaming, minimising delays and setbacks during gameplay. Within a domestic network, the regulations ought to be discerning and exceedingly uncomplicated. Utilising numerous disparate priorities may lead to unfavourable outcomes wherever none of the apps function adequately.

Bandwidth utilization

This configuration enables QoS-enabled applications to allocate a specific amount of network bandwidth. The standard allocation of network capacity for apps with QoS enabled is set at 80%. Figure 9 illustrates the potential utilisation of the currently unutilized bandwidth by other apps in the presence of QoS programmes.

Figure 9 Comparison of bandwidth utilization.

By employing a video conferencing application that makes use of high-priority bandwidth, we can monitor the utilisation of that particular resource. When QoS is activated, this programme is known to allocate 20% of the machine’s overall bandwidth, while the remaining 80% is left for all other network traffic. In addition to video conferencing, this technology offers a multitude of applications. Put simply, the criterion for dispatching packets is “first delivered packet first”. However, it is important to note that video conferencing app traffic will consistently be given priority over all other types of traffic. Additionally, it is mandated that no individual app is allowed to utilise more than 20% of the total available bandwidth.

Conclusion

The QoS applies priority encryption to the middle and presentation layers of the OSI model. This guarantees the prioritised delivery of the most crucial items. In the 5G OSI model, switches commonly function at the second layer, whereas routers typically operate at the third layer. Consequently, the switch will give preference to transmissions that exclusively utilise the 5G priority note, while network routers will ignore this information. QoS utilises a dedicated service protocol at the Transport Layer of the OSI model to give priority to data. The IP prefixes of packets sent via TCP/IP are altered to incorporate the QoS encoding. The proposed model achieved high levels of performance in several parameters, including QoS packet planner (92.8%), QoS packet scheduler (92.6%), standard packet selection (99.53%), traffic management (96.7%), maximum data rate (97.39%), and bandwidth utilisation (92.8%). Hence, the proposed model surpasses the existing models EESF, FADT, ERS and EMUA. As a result, QoS will enhance the network’s efficiency by reducing the impact of congestion and rearranging traffic patterns to accelerate the transfer of high-priority data. Future research directions could explore further refinement of the MCM model, possibly by incorporating more advanced machine learning techniques for real-time data analytics and network condition forecasting.

Supplemental Information

Supplemental Information 1 Experimental results.

Supplemental Information 2 Code.

Additional Information and Declarations

Competing Interests

Author Contributions

Data Availability

The authors declare that they have no competing interests.

Abdullah M. Alashjaee performed the experiments, performed the computation work, prepared figures and/or tables, and approved the final draft.

Sumit Kushwaha conceived and designed the experiments, performed the experiments, analyzed the data, performed the computation work, prepared figures and/or tables, authored or reviewed drafts of the article, and approved the final draft.

Hayam Alamro performed the experiments, authored or reviewed drafts of the article, and approved the final draft.

Asma Abbas Hassan performed the experiments, authored or reviewed drafts of the article, and approved the final draft.

Fuhid Alanazi analyzed the data, authored or reviewed drafts of the article, and approved the final draft.

Abdullah Mohamed analyzed the data, authored or reviewed drafts of the article, and approved the final draft.

The following information was supplied regarding data availability:

The raw data and code are available in the Supplemental Files and at GitHub: https://github.com/Dr-Sumit-Kushwaha/Optimizing-5G-Network-Performance-with-Dynamic-Resource-Allocation-Robust-Encryption.

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
