# Peer review of "Optimizing 5G network performance with dynamic resource allocation, robust encryption and Quality of Service (QoS) enhancement"

_PeerJ Computer Science, doi:10.7717/peerj-cs.2567_

## Round 0.1 · original submission · Major Revisions

Dear authors,

Thank you for submitting your article. Feedback from the reviewers is now available. It is not recommended that your article be published in its current format. However, we strongly recommend that you address the issues raised by the reviewers, especially those related to readability, experimental design and validity, and resubmit your paper after making the necessary changes. Before submitting the paper following should also be addressed:

1. Although the title contains “optimization”, there is not any clear optimization process in the manuscript. Optimization modelling is absent. Decision variables (types, boundaries etc.), objective function, constraint functions, etc. are not provided
2. Please write research gap and the motivation of the study. Evaluate how your study is different from others. Please highlight the originality and advantages of the proposed method.
3. More recent literature should be examined.
4. In the conclusions, please state explicitly what lessons can be learnt from this study and then describe in more detail the future research directions.

Best wishes,

·

Basic reporting

The paper proposes to optimize 5g network performance through the use of the MCM algorithm. The paper introduces the 5G network environment supported by recent references. The authors used good English when writing their research paper. It is easy to read and understand what the authors wished to achieve as they used professional English throughout the paper.

The authors used recent literature references to support their work and provided sufficient background in the introduction section. However, the paper lacks a related work section that discusses what other researchers have proposed previously to solve a similar problem, authors need to discuss previously proposed solutions including what was done, why, how, results, and limitations of each solution. This is very important as it will be used to support some of the techniques selected to be compared with the proposed algorithm.

The paper used a professional article structure and figures. However, the authors did not discuss the results in details as to why these algorithms compared performed the way they performed. This is very important and the authors should address this.

Experimental design

It is not clear how the proposed algorithm was designed and implemented to solve the identified problem. It is not clear which techniques did the authors use when designing the proposed algorithm. This affected the way they discussed the results as they couldn't support why the proposed algorithm performed the way it performed.

Validity of the findings

It is very challenging to confirm the validity of the results as it is not clear how the proposed algorithm was designed and implemented. It is not mentioned which tools were used to simulator the proposed algorithm. How many experiments were run to get the results presented.

Additional comments

Authors should provide a summary of the results in the abstract as they did in the conclusion section.

IEEE references should be in order of appearance meaning the reference numbers on line 76 should be [12], [13]. This is very important.

Authors should provide the full name of each acronym when used for the first time. Thereafter, they should use the acronym as they continue. Check lines 113, 124, and 182, just to point out a few errors from the paper.

Before Section 2, add a summary that presents what each Section of the paper will present.

Line 185 is incomplete, what do you mean by others while you only mentioned one protocol?

The sentences in lines 203 and 204 mean one thing, please avoid repetitions.

Section 2 should be rewritten. The related work section should discuss what was done, why, how, result, and limitations to support the need for your proposed algorithm. This is very important as it is the core of the paper.

The info presented under Figure 1 is too general. The authors do not say anything about the proposed algorithm and the algorithms compared with it.

Algorithm 1 --- What are your inputs? It is not clear how the proposed algorithm will solve the identified problem. Which techniques were integrated when designing the proposed algorithm?

In section 4, the authors need to support why the algorithms selected to be compared with the proposed algorithm were selected. This could have also been achieved through the discussion in the related work section.

The authors did not say things as to what contributed to the results presented in Figures 4 to 9. This is very important as it will show that the authors understand if they have achieved what they wanted to achieve and how they have achieved it.

The references on the reference list are up to 33 but only 28 references were in-texted referenced.

Figure 1 - Which techniques are used for encryption and decryption so that QoS is improved? You cannot just mention encryption and decryption and then leave it there. How does the proposed algorithm use Figure 1 to reduce latency?

Figure 2 - Which one is stage what?

Figure 3 - The inputs presented in Figure 3 are not appearing on Algorithm 1.

Reviewer 2 ·

Basic reporting

The authors present 5G Network Performance with Dynamic Resource Allocation, Robust Encryption and QoS Enhancement. The study is interesting. In general, the main conclusions presented in the paper are supported by the figures and supporting text. However, to meet the journal quality standards, the following comments need to be addressed.
• Abstract: Should be improved and extended. The authors talk lot about the problem formulation, but novelty of the proposed model is missing. Also provided the general applicability of their model. Please be specific what are the main quantitative results to attract general audiences.
• The introduction can be improved. The authors should focus on extending the novelty of the current study. Emphasize should be given in improvement of the model (in quantitative sense) compared to existing state-of-the art models.
• More details about network architecture and complexity of the model should be provided.
• what about comparison of the result with current state-of-the art models? Did authors perform ablation study to compare with different models?
• What are the baseline models and benchmark results? The authors may compared the result with existing models evaluated with datasets
• Conclusion parts needs to be strengthened.
• Please provide a fair weakness and limitation of the model, and how it can be improved.
• Typographical errors: There are several minor grammatical errors and incorrect sentence structures. Please run this through a spell checker.
Discussions of relevant literature could be further enhanced, which can help better motivate the current study and link to the existing work. Authors should consider the following relevant recent works in the field of applying SOTA deep learning techniques to better motivate the usefulness of machine learning approaches, such as
see :- Neural Networks 2022 https://doi.org/10.1016/j.neunet.2022.05.024

Hence they should be briefly discussed in the related work section.

Experimental design

see section 1

Validity of the findings

see section 1

Additional comments

see section 1

---

## Round 0.2 · accepted · Accept

Dear Authors,

I am grateful for your efforts in revising the paper. The invitation to review the previous version was not responded to in a timely manner by the appointed reviewers. However, I have assessed the revised manuscript myself, and am satisfied with the changes made. I believe it is now ready for publication.

Best wishes,